# AgentXRay: White-Boxing Agentic Systems via Workflow Reconstruction

**Ruijie Shi** [1 2]  **Houbin Zhang** [1]  **Yuecheng Han** [3]  **Yuheng Wang** [3]  **Jingru Fan** [3]  **Runde Yang** [3]  **Yufan Dang** [1]
**Huatao Li** [3]  **Dewen Liu** [3]  **Yuan Cheng** [3]  **Chen Qian** [3]

## Abstract

Large Language Models have shown strong capabilities in complex problem solving, yet many agentic systems remain difficult to interpret and control due to opaque internal workflows. While some frameworks offer explicit architectures for collaboration, many deployed agentic systems operate as black boxes to users. We address this by introducing Agentic Workflow Reconstruction (AWR), a new task aiming to synthesize an explicit, interpretable stand-in workflow that approximates a black-box system using only input–output access. We propose AgentXRay, a search-based framework that formulates AWR as a combinatorial optimization problem over discrete agent roles and tool invocations in a chain-structured workflow space. Unlike model distillation, AgentXRay produces editable white-box workflows that match target outputs under an observable, output-based proxy metric, without accessing model parameters. To navigate the vast search space, AgentXRay employs Monte Carlo Tree Search enhanced by a scoring-based Red-Black Pruning mechanism, which dynamically integrates proxy quality with search depth. Experiments across diverse domains demonstrate that AgentXRay achieves higher proxy similarity and reduces token consumption compared to unpruned search, enabling deeper workflow exploration under fixed iteration budgets.

## 1. Introduction

Large Language Models (LLMs) have exhibited remarkable capabilities in complex problem-solving, yet they continue to face bottlenecks stemming from their reliance on internal-

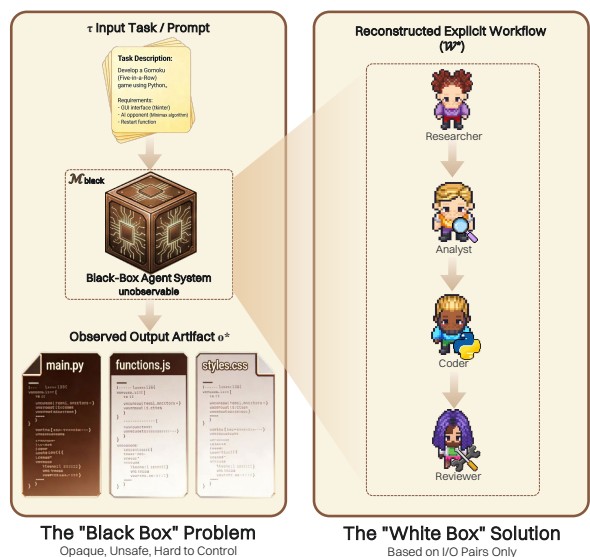

*Figure 1.* The concept of AWR. Given a black-box system $\mathcal{M}_{\text{black}}$ producing output $o^*$ from input $\tau$, the goal is to synthesize an explicit, interpretable white-box workflow $\mathcal{W}^*$ (e.g., a sequence of specialized agents) that matches the target outputs under observable outputs, using only input-output pairs.

ized parameters, which limits access to real-time information and specialized tools (Zhang et al., 2025b; Lewis et al., 2020; Kandpal et al., 2023). To transcend these limitations, standalone models have evolved into agents by incorporating dynamic planning, tool usage, and long-term memory (Xi et al., 2025; Liu et al., 2024b; Yao et al., 2023b; Schick et al., 2023; Wang et al., 2023a; Qin et al., 2023; Park et al., 2023; Shen et al., 2023) to expand their execution boundaries. Along this line, Multi-Agent Systems (MAS) leverage collective intelligence through function specialization and structured collaboration (Li et al., 2023a; Wu et al., 2024), effectively addressing intricate tasks that demand multi-step reasoning and cross-domain expertise.

However, existing high-performance systems often operate as "black boxes": their internal mechanisms—from sequential tool patterns to complex multi-agent topologies—remain opaque. The complexity of interactions further exacerbates this lack of interpretability (Domenech i Vila et al., 2024). Explaining agent behaviors is crucial for trustworthy AI, yet opacity remains a significant hurdle (Gimenez-Abalos

[1]Tsinghua University, Beijing, China [2]Massachusetts Institute of Technology, Cambridge, MA, USA [3]Shanghai Jiao Tong University, Shanghai, China. Correspondence to: Ruijie Shi <rjshi@mit.edu>, Chen Qian <qianc@sjtu.edu.cn>.

*Proceedings of the 43rd International Conference on Machine Learning*, Seoul, South Korea. PMLR 306, 2026. Copyright 2026 by the author(s).

et al., 2024). Consequently, users struggle to comprehend decision-making processes, which limits adaptation to downstream tasks and hinders deployment in safety-critical domains (Salehi et al., 2025; Amodei et al., 2016; Bommasani et al., 2021; Hendrycks et al., 2021).

Motivated by this, we aim to build interpretable surrogate workflows (i.e., stand-in workflows) to improve transparency and controllability. We formally introduce **Agentic Workflow Reconstruction (AWR)**: given a dataset of Input-Output pairs collected from a black-box system, the goal is to synthesize an explicit white-box *surrogate* workflow (a stand-in workflow) that approximates the target behavior under an observable, output-based proxy metric.

We formulate AWR as a combinatorial optimization problem to identify the optimal configuration of agents and tools (Hu et al., 2025; Zhang et al., 2025a; 2024). Given the vast search space, we propose **AgentXRay**. We adopt the **linearity hypothesis** (Qian et al., 2025; Dang et al., 2025) to represent workflows as sequential trajectories, casting reconstruction as a search for high-scoring agentic primitives. To navigate this space, we employ **Monte Carlo Tree Search (MCTS)** (Yao et al., 2023a; Zhou et al., 2024), which effectively handles delayed rewards and balances exploration with exploitation. To mitigate the combinatorial explosion (Zhang et al., 2024), we introduce a specialized **"Red-Black" Pruning mechanism**. This mechanism filters low-potential branches based on a multi-dimensional scoring function, concentrating resources on promising candidates. The resulting workflow serves as an interpretable proxy for the target system, enabling cost-effective adaptation.

We validate our framework across five domains: software development (ChatDev), data analysis (MetaGPT), education (TeachMaster), 3D modeling (ChatGPT), and scientific computing (Gemini). Empirical results demonstrate that AgentXRay recovers workflows with high *proxy* similarity (Static Functional Equivalence; avg. 0.426) and significantly improves search efficiency (8–22% token reduction) compared to unpruned baselines.

Our main contributions are:

1. **Task.** We introduce **AWR**, a new task that converts *black-box* LLM agent systems into *white-box* counterparts by reconstructing *explicit workflows* from input–output observations.

2. **Method.** We propose **AgentXRay**, an MCTS-based search framework equipped with a specialized **Red-Black Pruning** mechanism, which mitigates combinatorial explosion in the primitive space and enables practical reconstruction with improved search efficiency.

3. **Evaluation.** We validate AgentXRay across five domains (*code generation, data analysis, education, sci-*

*entific computing, 3D modeling*) by reconstructing representative systems (e.g., **ChatDev** (Qian et al., 2024), **MetaGPT** (Hong et al., 2023), **TeachMaster** (Wang et al., 2026), **ChatGPT (GPT-5.2)**, and **Gemini 3 Pro**, both accessed via their public APIs).

**Conflict of Interest Disclosure.** The authors declare no financial conflicts of interest. The black-box agentic systems evaluated in this work (e.g., ChatGPT, Gemini) and the LLMs used to instantiate AgentXRay's primitive space (e.g., GPT-4, Claude, DeepSeek, Qwen, GLM) are not developed by any institution that employs the authors.

## 2. Method

### 2.1. Problem Formulation

We study *AWR*: constructing an interpretable surrogate workflow that matches a black-box system's *final outputs* from input-output observations.

**Task Definition and Inputs.** Let $\mathcal{M}_{\text{black}} : \mathcal{X} \to \mathcal{Y}$ denote a black-box agentic system (including both multi-agent and complex single-agent workflows), where internal components (e.g., agent roles, prompts, and coordination topology) are hidden and only the I-O interface is accessible. We are given a dataset $\mathcal{D} = \{(\tau_i, o_i^*)\}_{i=1}^N$, where $\tau_i$ is a task description and $o_i^* = \mathcal{M}_{\text{black}}(\tau_i)$ is the corresponding output. Our goal is to synthesize a white-box workflow $\mathcal{W}$ whose outputs match those of $\mathcal{M}_{\text{black}}$ under an evaluation metric.

**Unified Primitive Space.** We unify agents and tools into a single primitive space $\Omega$ of *agentic primitives*. Each primitive is a tuple $p = \langle \rho, \mu, \pi, T_{\text{local}} \rangle$, where $\rho$ is the role, $\mu$ is the base model, $\pi$ is the thought pattern, and $T_{\text{local}}$ is the attached toolset. This definition treats both pure-reasoning agents ($T_{\text{local}} = \emptyset$) and tool-augmented agents ($T_{\text{local}} \neq \emptyset$) as atomic search units, consistent with tool-augmented LMs and API-based reasoning (Patil et al., 2024; Tang et al., 2023; Lu et al., 2023; Paranjape et al., 2023; Qin et al., 2023; Li et al., 2023b; Song et al., 2023).

**Workflow Representation and Linearity.** We represent a workflow $\mathcal{W}$ as a linear sequence of length at most $L_{\text{max}}$: $\mathbf{s} = [s_1, s_2, \ldots, s_L]$ with $s_j \in \Omega$ and $1 \leq L \leq L_{\text{max}}$. Searching arbitrary graph topologies is computationally prohibitive ($O(2^{|\Omega|^2})$). We therefore adopt the *Linearity Hypothesis* and restrict reconstruction to sequential workflows that capture the *execution-time ordering* of agent/tool calls.

This restriction is well-motivated by how many agentic systems are executed in practice. First, Qian et al. (2025) (MacNet) organizes multi-agent collaboration with DAG structures, but the system still executes agents in a *topological order*, which induces a concrete, time-ordered trace. Second, Dang et al. (2025) show that when an orchestrator

*Figure 2.* Overview of the AgentXRay framework. The process takes task inputs and black-box outputs, searches for a high-scoring primitive sequence via MCTS with Red-Black Pruning, and returns an interpretable white-box workflow.

is trained to dynamically coordinate agents without explicit topological constraints, effective coordination can still be realized as a compact chain-like policy under efficiency pressures. Moreover, many deployed LLM agents interact with their environment through an iterative *action–observation* loop, yielding a trajectory that is naturally linearizable; for example, ReAct interleaves reasoning with stepwise actions conditioned on intermediate observations, and web agents evaluated in realistic environments such as WebArena operate as ordered sequences of atomic actions with feedback at each step (Yao et al., 2023b; Zhou et al., 2023).

These observations suggest that (i) complex coordination often *serializes* into an execution-time ordering even when the design is graph-structured, and (ii) sequential traces are the dominant observable object under strict black-box access. Our linearity hypothesis is therefore an execution-oriented surrogate that keeps reconstruction tractable while aligning with how agent systems behave at runtime.

**Optimization Objective.** We seek a workflow that maximizes the expected similarity between reconstructed outputs and black-box outputs on $\mathcal{D}$:

$$\mathbf{s}^* = \underset{\mathbf{s} \in \Omega^{\leq L_{\max}}}{\arg\max} \, \mathbb{E}_{(\tau, o^*) \sim \mathcal{D}} \left[ \mathrm{Sim}(\Phi(\mathbf{s}, \tau), o^*) \right], \quad (1)$$

where $\Phi(\mathbf{s}, \tau)$ denotes executing the workflow $\mathbf{s}$ on input $\tau$, and $\mathrm{Sim}(\cdot, \cdot) \in [0, 1]$ is a task-specific semantic/functional metric (implemented by an external evaluator; e.g., AST-based matching for code or cosine similarity for text).

**Behavioral Fidelity as the Reconstruction Goal.** Our ob-

jective is to reconstruct a workflow whose **input–output behavior** matches the target system: for the same input, the reconstructed system should produce an output as similar as possible to the target output, regardless of internal traces or implementation choices. In open-ended, multi-file settings, directly verifying true functional equivalence can be infeasible (non-portable environments, external tools/services, stochastic APIs, safety constraints). We therefore optimize *behavioral fidelity* using scalable **proxy** metrics (e.g., SFE) that approximate output-level consistency at scale.

### 2.2. AgentXRay

We propose *AgentXRay*, an MCTS-based search procedure over workflow prefixes.

**MCTS States and Actions.** Each tree node $v$ represents a partial workflow prefix $s_{1:t}$ (the root is the empty prefix), and an edge corresponds to appending one primitive $p \in \Omega$. This view applies uniformly to multi-agent and single-agent settings: the search composes an explicit sequence of role/tool primitives to form $\mathbf{s}$. We use MCTS (Kocsis & Szepesvári, 2006; Auer et al., 2002; Browne et al., 2012) to address delayed feedback: $\mathrm{Sim}(\cdot)$ is only observable after executing (nearly) complete workflows. UCB (Auer et al., 2002) balances exploration and exploitation over a large heterogeneous action space. To control branching, AgentXRay maintains a *dynamic Red-Black coloring* that guides whether the search prioritizes depth refinement or width expansion.

**Algorithm 1** AgentXRay: MCTS with Red-Black Pruning

1: **Input:** Primitive space $\Omega$, dataset $\mathcal{D}$, iterations $N$
2: **Output:** Reconstructed workflow $\mathcal{W}^*$
3: $root \leftarrow \text{NODE}(\text{depth} = 0)$
4: **for** $t = 1$ **to** $N$ **do**
5: $\quad (\tau, o^*) \leftarrow \mathcal{D}[t \bmod |\mathcal{D}|]$
6: $\quad \text{COLORTREE}(root, \beta)$ {Dynamic coloring}
7: $\quad v \leftarrow root; content \leftarrow$ ""
8: $\quad$ **while** $\neg v.terminal$ **do**
9: $\quad\quad v' \leftarrow \text{SELECTCHILD}(v, \tau, \Omega)$ {Color-guided}
10: $\quad\quad resp \leftarrow v'.\text{EXECUTE}(\tau, content)$
11: $\quad\quad$ **if** $resp = \text{FAIL}$ **then**
12: $\quad\quad\quad v'.terminal \leftarrow \text{TRUE}; r \leftarrow 0;$ **break**
13: $\quad\quad$ **end if**
14: $\quad\quad v'.content \leftarrow resp$
15: $\quad\quad$ **if** $\neg v'.expanded$ **then**
16: $\quad\quad\quad r \leftarrow \text{SIMULATE}(v', \tau, o^*)$ {Rollout}
17: $\quad\quad\quad v \leftarrow v';$ **break**
18: $\quad\quad$ **end if**
19: $\quad\quad r \leftarrow \text{SIM}(\text{EXTRACTCODE}(resp), o^*)$
20: $\quad\quad content \leftarrow resp; v \leftarrow v'$
21: $\quad$ **end while**
22: $\quad \text{BACKUP}(v, r)$ {Backpropagate reward}
23: **end for**
24: **return** $\text{BESTPATH}(root)$

## 2.2.1. SEARCH CYCLE

Each iteration performs selection/expansion, simulation, and backup:

**1. Color-Guided Selection/Expansion.** Let $v$ be the current node. If $v$ is RED, we treat its current choice as stabilized and select among existing children using UCB to continue descending (depth refinement). If $v$ is BLACK, we prioritize width exploration by creating a new child of $v$ (branching), thus increasing the number of $v$'s children.

**2. Simulation (Rollout).** For a newly expanded node, we complete the workflow by sampling primitives until reaching length $L_{\max}$, execute the resulting workflow on a sampled task $\tau$, and obtain output $o$. We use early stopping: if execution fails (e.g., primitive error or generation failure), we assign reward $r = 0$. Otherwise, $r = \text{Sim}(o, o^*)$.

**3. Backup.** We backpropagate $r$ along the visited path, updating $N(v) \leftarrow N(v) + 1$ and $Q(v) \leftarrow Q(v) + r$ for all ancestors.

Algorithm 1 summarizes the full search loop, integrating color-guided descent (Line 9), early-stopping rollouts (Lines 11–13), and reward backpropagation (Line 22) into a single procedure executed for $N$ iterations.

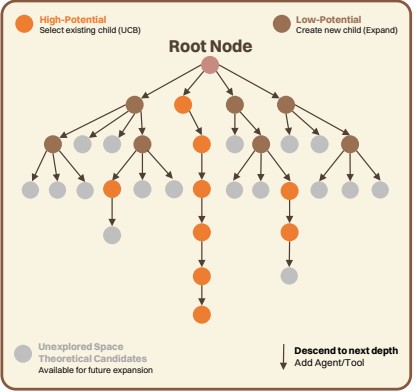

*Figure 3.* Dynamic Red-Black Pruning. Nodes are scored by Quality, Depth, and Width; high-scoring nodes (RED) select among existing children via UCB to refine depth, while low-scoring nodes (BLACK) expand by creating new children to broaden coverage. Gray nodes represent unexplored candidates.

## 2.2.2. DYNAMIC RED-BLACK COLORING AND PRUNING

To identify robust structural backbones during noisy early exploration, we assign each node a composite potential score based on *Quality*, *Depth*, and *Width* (Figure 3):

$$\text{Score}(v) = \underbrace{\frac{Q(v)}{N(v)}}_{\text{Quality}} \cdot \underbrace{\left(\frac{d(v) + 1}{L_{\max} + 1}\right)}_{\text{Depth}} \cdot \underbrace{\frac{|\mathcal{C}(v)|}{M}}_{\text{Width}}, \quad (2)$$

where $d(v)$ is the depth (prefix length), $|\mathcal{C}(v)|$ is the number of children, and $M$ is the maximum allowed number of children per node (a branching cap; in our implementation this corresponds to MAX_CHILDREN_NUMBER).

Let $\mathcal{L}_{\text{term}}$ denote the set of terminal nodes that should not be further expanded, including nodes that have reached depth $L_{\max}$ and nodes marked terminal due to execution failures during rollout (early stopping). At each iteration, we compute the $\beta$-quantile $\theta_\beta$ of $\text{Score}(\cdot)$ over active nodes and assign colors:

$$C(v) = \begin{cases} \text{RED}, & \text{if } \text{Score}(v) \geq \theta_\beta \ \wedge \ v \notin \mathcal{L}_{\text{term}}, \\ \text{BLACK}, & \text{otherwise.} \end{cases} \quad (3)$$

### 2.3. Theoretical Complexity Analysis

We analyze how Red-Black Pruning contracts the search space and yields an exponential speedup in depth. This analysis characterizes search-space contraction under an effective (realized) pruning rate and serves as an explanatory bound; it does not claim regret-optimality guarantees for the full algorithm.

Let $b = |\Omega|$ be the branching factor, $L_{\max}$ the maximum workflow length, and $p$ the realized pruning rate induced by the scoring-and-threshold rule. Here $p$ depends on the

score distribution and the quantile parameter $\beta$, while $\beta$ is a user-controlled threshold.

### 2.3.1. SEARCH SPACE CONTRACTION BOUNDS

**Unpruned Baseline.** Without pruning, the frontier size at depth $d$ is $N_d = b^d$, and the total search volume is

$$\mathcal{V}_{\text{full}} = \sum_{d=0}^{L_{\max}} b^d = \frac{b^{L_{\max}+1} - 1}{b - 1} = \Theta(b^{L_{\max}}). \quad (4)$$

**Lower Bound (Uniform Pruning).** Under pruning rate $p$, the effective frontier is $\hat{N}_d = (b(1-p))^d$, yielding

$$\mathcal{V}_{\text{eff}} = \sum_{d=0}^{L_{\max}} (b(1-p))^d = \Theta\big((b(1-p))^{L_{\max}}\big). \quad (5)$$

Thus the acceleration ratio satisfies

$$\eta(L_{\max}) = \frac{\mathcal{V}_{\text{full}}}{\mathcal{V}_{\text{eff}}} \geq \left(\frac{1}{1-p}\right)^{L_{\max}}. \quad (6)$$

**Upper Bound ($\beta$-Quantile Threshold).** In an idealized aggressive case, the rule retains at most a $(1-\beta)$ fraction of nodes per depth, giving $\mathcal{V}_{\min} = \Theta((b(1-\beta))^{L_{\max}})$ and

$$\eta(L_{\max}) \leq \left(\frac{1}{1-\beta}\right)^{L_{\max}}. \quad (7)$$

These bounds characterize search-space contraction in an idealized asymptotic regime. In practice, under strict iteration budgets ($N=20$ in our experiments), the realized speedup is moderate—empirically we observe 8–22% token reduction (Section 3.4)—because the search tree never grows large enough for the full asymptotic contraction to take effect. The practical value of pruning under finite budgets is therefore *budget allocation*: concentrating limited evaluations on high-potential backbones, which enables deeper workflows (Section 3.3) and improves final reconstruction quality.

## 3. Experiments

We evaluate AgentXRay along three axes: (i) whether the unified primitive space can express diverse agentic behaviors, (ii) whether MCTS can recover high-fidelity workflows under strict black-box access, and (iii) whether Red-Black Pruning improves budget efficiency. We reconstruct workflows on five open-ended, tool-augmented domains covering both open-source multi-agent frameworks (ChatDev, MetaGPT, TeachMaster) and proprietary assistants (Chat-GPT, Gemini), spanning code generation, data analysis, education, scientific computing, and 3D modeling.

### 3.1. Experimental Setup

**Tasks and Benchmarks.** We reconstruct five domains: (1) **Software Development**: ChatDev (Qian et al., 2024) on

52 multi-file Python generation tasks from SRDD. (2) **Data Analysis**: MetaGPT (Hong et al., 2023) on 52 MatPlot-Bench (Yang et al., 2024) visualization tasks. (3) **Education**: TeachMaster (Wang et al., 2026) on 25 automated teaching video generation tasks. (4) **3D Modeling**: Chat-GPT on 100 ScanRefer (Chen et al., 2020) scripting tasks. (5) **Scientific Computing**: Gemini on 80 SciBench (Wang et al., 2023b) problems. We treat all targets as strict black boxes and collect only input–output pairs. Our setup complements existing agent benchmarks that focus on white-box agent evaluation (Liu et al., 2023; Wang et al., 2023c; Shridhar et al., 2020) by instead asking whether agentic behavior can be *recovered* from outputs alone.

**Baselines.** We compare AgentXRay with four categories of methods that test different hypotheses: (1) **SFT**: supervised fine-tuning on input–output pairs (Touvron et al., 2023), testing whether behavior cloning can capture agentic workflows; (2) **Claude Opus 4.5**: a strong single-model baseline with multi-turn self-refinement (Madaan et al., 2023), testing whether raw model capacity can substitute for structured search; (3) **ReAct (Claude Opus 4.5)**: Claude Opus 4.5 with ReAct-style tool use (Yao et al., 2023b), testing whether tool augmentation alone bridges the gap; (4) **AFlow**: MCTS-based workflow search without Red-Black Pruning (Zhang et al., 2025a), isolating our pruning contribution.

Notably, the Claude baselines use a stronger model than any component in AgentXRay's primitive space. To reduce prompt sensitivity for the strong single-model baselines, we evaluate a small set of prompt variants and report the best-performing variant under the fixed interaction budget.

**Budget Parity Protocol.** We fix a max number of interaction rounds for every method, where one round equals one model invocation (including any internal self-refinement). All methods share the same tool set and per-round tool limits; token usage is measured and reported post hoc.

**Implementation Details.** We leverage both proprietary and open-weight LLMs to construct the primitive space, and access all models via API endpoints. Proprietary models include `gpt-4-turbo`, `gpt-4`, `gpt-4o-mini`, and `gpt-3.5-turbo` (Achiam et al., 2023). We additionally include strong third-party models served via compatible APIs, including `claude-sonnet-4-20250514`. Open-weight models include `llama-4-maverick-17b-128e-instruct` (Dubey et al., 2024), `deepseek-v3.2` (Liu et al., 2024a), `qwen3-coder-30b-a3b-instruct` (Yang et al., 2025), and `glm-4.7` (Zeng et al., 2022).

**Evaluation Metric.** We measure *proxy fidelity*: given the same input, does the reconstructed workflow produce an output artifact similar to the target system, regardless of unobservable internal traces. Because targets are strict

*Table 1.* Reconstruction performance comparison across five domains. We report similarity scores where higher is better. **Bold** indicates the best result, and underlined indicates the second best. Subscript arrows indicate the performance gap compared to the strong baseline AFLOW ($\uparrow$ improvement, $\downarrow$ decline). $\dagger$ denotes a statistically significant difference ($p \leq 0.05$) between a method and the strong baseline AFLOW.

| METHOD | CHATDEV | METAGPT | TEACHMASTER | 3D MODELING | SCI | AVG |
|---|---|---|---|---|---|---|
| SFT | $0.355^{\dagger}_{\downarrow 0.048}$ | $0.272_{\downarrow 0.008}$ | $0.124^{\dagger}_{\downarrow 0.224}$ | $0.091^{\dagger}_{\downarrow 0.199}$ | $0.139^{\dagger}_{\downarrow 0.234}$ | $0.196^{\dagger}$ |
| CLAUDE | $0.256^{\dagger}_{\downarrow 0.147}$ | $0.322_{\downarrow 0.042}$ | $0.303^{\dagger}_{\downarrow 0.045}$ | $0.282_{\downarrow 0.008}$ | $0.299^{\dagger}_{\downarrow 0.074}$ | $0.292^{\dagger}$ |
| REACT | $0.267^{\dagger}_{\downarrow 0.136}$ | $0.331_{\uparrow 0.051}$ | $0.322_{\downarrow 0.026}$ | $0.270_{\downarrow 0.020}$ | $0.305^{\dagger}_{\downarrow 0.068}$ | $0.299^{\dagger}$ |
| AFLOW | $0.403$ | $0.280$ | $0.348$ | $0.290$ | $0.373$ | $0.339$ |
| AGENTXRAY w/o Tools | $0.413_{\uparrow 0.010}$ | $0.301^{\dagger}_{\uparrow 0.021}$ | $0.357^{\dagger}_{\uparrow 0.009}$ | $\underline{0.332}^{\dagger}_{\uparrow 0.042}$ | $0.378^{\dagger}_{\uparrow 0.005}$ | $0.356^{\dagger}$ |
| AGENTXRAY w/o Pruning | $0.286^{\dagger}_{\downarrow 0.117}$ | $0.334_{\uparrow 0.054}$ | $0.378_{\uparrow 0.030}$ | $0.279_{\downarrow 0.011}$ | $0.312^{\dagger}_{\downarrow 0.061}$ | $0.318$ |
| AGENTXRAY (Selected) | $\mathbf{0.509}^{\dagger}_{\uparrow 0.106}$ | $\underline{0.470}^{\dagger}_{\uparrow 0.190}$ | $\mathbf{0.399}^{\dagger}_{\uparrow 0.051}$ | $0.318_{\uparrow 0.028}$ | $\mathbf{0.407}^{\dagger}_{\uparrow 0.034}$ | $\underline{0.421}^{\dagger}$ |
| AGENTXRAY (All Tools) | $\underline{0.425}^{\dagger}_{\uparrow 0.022}$ | $\mathbf{0.557}^{\dagger}_{\uparrow 0.277}$ | $\underline{0.390}^{\dagger}_{\uparrow 0.042}$ | $\mathbf{0.362}^{\dagger}_{\uparrow 0.072}$ | $\underline{0.395}^{\dagger}_{\uparrow 0.022}$ | $\mathbf{0.426}^{\dagger}$ |

black boxes and executing open-ended multi-file outputs can be costly or non-portable, executable-test-based evaluation (Austin et al., 2021; Chen et al., 2021) is impractical at scale. We therefore adopt **Static Functional Equivalence (SFE)** (Ren et al., 2020)—a structure-aware code similarity metric—as our scalable proxy. Concretely, SFE combines three AST-derived components with fixed weights: *interface similarity* (20%; function/class signatures and module structure), *logic similarity* (50%; algorithmic patterns, control flow, and computational expressions), and *semantic similarity* (30%; identifier semantics with synonym normalization, plus code-intent features from comments and docstrings). The logic-dominant weighting reflects that algorithmic structure is most indicative of behavioral equivalence in code-centric domains, while interface and naming provide complementary structural and intent signals. We use a TF–IDF cosine fallback (Reimers & Gurevych, 2019; Zhang et al., 2019) when AST parsing fails ($<$5% of samples).

**Human validation of SFE.** To validate SFE as a proxy for *output-level* similarity, we conduct a small-scale blind human study with three annotators on 30 stratified-randomly sampled output pairs across domains and SFE score ranges. Human similarity scores (0–1) are positively correlated with SFE (Spearman $\rho$=0.61, $p$<0.001, 95% CI [0.30, 0.80]); inter-annotator agreement is moderate (Krippendorff's $\alpha$=0.57).

### 3.2. Main Results

Table 1 reports the reconstruction performance across five domains. We organize our analysis around the hypotheses tested by each baseline category.

**Overall Performance.** AgentXRay achieves the best reconstruction fidelity across all five domains, with an average SFE of **0.426** (vs. AFlow 0.339 and Claude Opus 4.5 + ReAct 0.299) under the same interaction budget. Gains are

consistent across heterogeneous targets, indicating that explicit workflow search over agentic primitives is a robust approach to AWR.

**Can Behavior Cloning Suffice?** SFT performs worst (avg. 0.196), especially on Education and 3D, suggesting that direct input$\rightarrow$output cloning is inadequate for agentic systems. The key limitation is missing supervision on intermediate decomposition and tool-logic: input-output pairs do not reveal the underlying primitive sequence, so SFT may match surface form but fails to recover procedural structure (tool calls, control flow, coordination) (Pomerleau, 1991; Ross et al., 2011; De Haan et al., 2019; Nagarajan et al., 2020).

**Can Model Capacity Compensate for Lack of Structure?** Even a stronger single model (Claude Opus 4.5) reaches 0.292, and adding ReAct yields only 0.299 under the shared budget, indicating that capacity alone does not reliably discover compositional workflows. In contrast, AgentXRay's structured search explores and selects higher-fidelity workflow candidates within the same budget.

**Does Pruning Matter?** AgentXRay substantially outperforms AFlow (0.426 vs. 0.339 avg.), while removing pruning can even underperform AFlow (e.g., ChatDev: 0.286). This is because our unified primitive space increases the branching factor; without pruning, MCTS disperses the limited budget over many low-quality branches. Red-Black Pruning concentrates rollouts on high-potential prefixes, enabling deeper, coherent workflows under fixed $N$.

**Cross-Domain Patterns.** MetaGPT is most reconstructable (0.557), consistent with recurring analysis workflows well-covered by our primitive space, while ChatDev exhibits higher variance due to multi-file dependency sensitivity. Proprietary targets remain moderately reconstructable, supporting the feasibility of AWR under strict black-box access.

*Table 2.* Impact of Red-Black Pruning on achieved search depth. Red% denotes the proportion of nodes marked for deep exploration; Len denotes the final workflow length. Without pruning, search stagnates at $L = 2$; with pruning, search reaches deeper workflows.

| METHOD | CHATDEV | | METAGPT | | TEACHMASTER | | 3D | | SCI | |
|---|---|---|---|---|---|---|---|---|---|---|
| | RED% | LEN | RED% | LEN | RED% | LEN | RED% | LEN | RED% | LEN |
| W/O PRUNING | 0 | 2 | 0 | 2 | 0 | 2 | 0 | 2 | 0 | 2 |
| W/O TOOLS | 55 | 6 | 55 | 2 | 55 | 4 | 65 | 6 | 45 | 6 |
| ALL TOOLS | 55 | 6 | 55 | 2 | 60 | 5 | 55 | 2 | 55 | 6 |
| SEL. TOOLS | 45 | **6** | 50 | 4 | 45 | **6** | 60 | **6** | 65 | **6** |

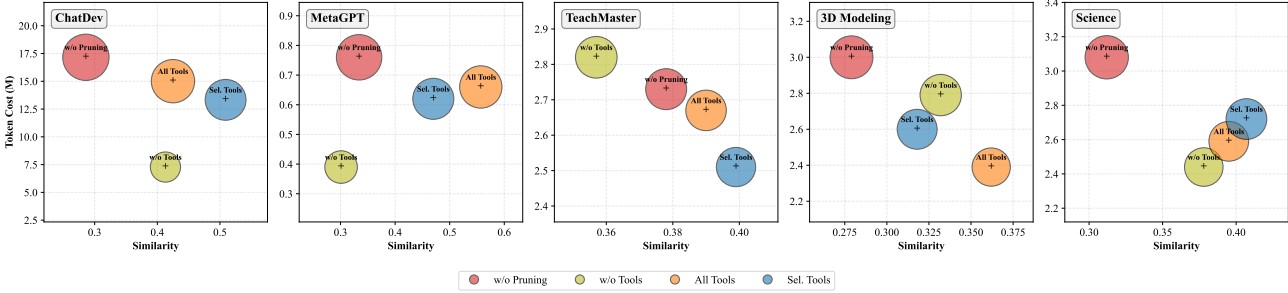

*Figure 4.* Cost-Efficiency analysis across five domains. The horizontal axis denotes reconstruction similarity (higher is better), while the vertical axis represents token consumption in millions (lower is better). Our method achieves comparable or higher fidelity than unpruned variants but with reduced computational overhead.

### 3.3. Ablation Study

We conduct systematic ablations to isolate the contribution of each component.

**Tool Integration.** Removing tools decreases average performance from 0.426 to 0.356 (−16.4%), with domain-dependent impact: substantial for MetaGPT (−46.0%) but marginal for Scientific Computing (−4.3%), reflecting whether the target system relies on external tool execution.

**Pruning Enables Deeper Search.** Table 2 shows that without pruning, MCTS stagnates at shallow depth ($L = 2$) as the budget dissipates on the exponential frontier. With Red-Black Pruning, search reaches maximum depth ($L = 6$). The Red node proportions (45–65%) confirm that pruning concentrates budget on promising workflows, explaining AgentXRay's superior performance.

### 3.4. Efficiency Analysis

Having validated reconstruction quality, we now turn to our third claim: Red-Black Pruning significantly improves search efficiency. This is critical for practical deployment, as the combinatorial search space would be intractable without effective pruning.

**Token Consumption.** Figure 4 illustrates the cost-efficiency trade-off across five domains. Comparing "AgentXRay (Sel. Tools)" with "AgentXRay w/o Pruning", the Red-Black Pruning mechanism reduces token consumption by **8% to 22%** across all five domains, with the largest reduction on

ChatDev (22.3%, from 17.16M to 13.33M tokens). The "w/o Tools" variant consumes the fewest tokens by avoiding tool execution overhead, but this comes at the cost of reconstruction quality (0.356 vs. 0.426 average, per Table 1). Our full method achieves a favorable balance: high reconstruction fidelity with substantially reduced cost.

The domain-specific patterns are also informative. ChatDev exhibits the highest consumption because multi-file code generation requires expensive repeated evaluation for each workflow candidate. MetaGPT is the most efficient due to shorter interaction cycles and lighter verification. The remaining three domains cluster around 2.5–3.0M tokens, suggesting comparable evaluation overhead despite their diverse application areas.

**Convergence Behavior.** Figure 5 illustrates how reconstruction quality evolves with cumulative token consumption on two representative domains. Two key patterns emerge: (1) *Earlier convergence*—the pruned variants ("Sel. Tools" and "All Tools") reach high-quality solutions within fewer tokens, while "w/o Pruning" requires substantially more budget to achieve comparable scores; (2) *Steeper ascent*—pruning concentrates exploration on promising branches, leading to faster quality improvement per token spent. For example, on ChatDev, "Sel. Tools" achieves 0.65+ score at ∼10M tokens, whereas "w/o Pruning" requires ∼12M tokens to reach the same level. These curves reflect training performance during MCTS; final test scores (Table 1) are evaluated separately.

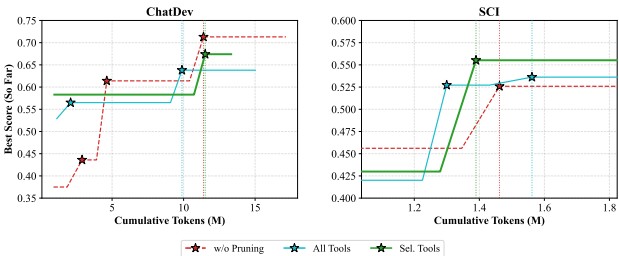

*Figure 5.* Convergence analysis on ChatDev and SCI. Pruned variants converge earlier and faster, reaching strong candidates at lower token cost than unpruned search.

**Practical Considerations.** Our complexity analysis characterizes search-space contraction in an idealized asymptotic regime; the empirical token savings (8–22%) are smaller but consistent with the qualitative prediction that pruning enables deeper, more focused exploration. Token consumption is measured uniformly for AgentXRay and all baselines under the same interaction budget, including strong single-model baselines with self-refinement.

**Statistical Robustness.** To assess reproducibility despite API-based non-determinism, we conducted 5-seed runs across representative domains and performed paired t-tests against AFlow. Results marked with [†] in Table 1 indicate statistically significant improvements ($p < 0.05$). Key findings: (1) All "AgentXRay" variants with pruning achieve significant improvements over AFlow on average. (2) The relative ranking of methods is preserved across all seeds. (3) Even worst-case runs exceed AFlow, confirming that improvements are not artifacts of favorable seed selection. These results demonstrate that our methodology is robust to the stochasticity inherent in LLM API calls.

### 3.5. Generalization to Open-Source Models

A natural question arises: *does AgentXRay depend on proprietary models?* We conduct an additional experiment targeting **Atoms**[1], a multi-agent platform evolved from MetaGPT. We use the Automatic Slide Generation dataset (Sefid et al., 2021) containing 80 presentation generation tasks, and construct the primitive space using *exclusively open-weight models*.

*Table 3.* Reconstruction on Atoms using open-weight models.

| METHOD | SIMILARITY |
|---|---|
| SFT | 0.275 |
| AFLOW | 0.321 |
| AGENTXRAY W/O TOOLS | 0.275 |
| AGENTXRAY (SELECTED TOOLS) | 0.326 |
| AGENTXRAY W/O PRUNING | 0.328 |
| AGENTXRAY (ALL TOOLS) | **0.337** |

AgentXRay (All Tools) achieves the best performance

---

[1] https://atoms.dev

(0.337), outperforming AFlow by 5.0%. The ablation trends mirror earlier domains, suggesting that AgentXRay's effectiveness stems from its algorithmic design rather than privileged access to proprietary models.

## 4. Discussion

We formally introduce AWR as a research task: constructing an editable, interpretable *surrogate workflow* from black-box input–output observations, with the goal of matching the target system's *final outputs* rather than recovering hidden internals. Unlike *workflow generation* (e.g., AFlow) that optimizes from scratch, or *model distillation* that transfers behavior into parameters, AWR yields explicit workflows—offering a complementary path for understanding, debugging, and reusing high-performing systems when internal designs are unavailable.

While AgentXRay demonstrates the feasibility of AWR, several limitations and future directions remain. Conceptually, AWR is closely related in spirit to model extraction via prediction APIs (Tramèr et al., 2016), where black-box query access can be used to duplicate functionality; our setting differs in that we recover an *editable workflow* rather than a parametric replica.

**Search Efficiency.** Our empirical efficiency gains (8–22% token reduction) are smaller than asymptotic bounds because we operate under strict budgets ($N = 20$). The main value of Red-Black Pruning is *budget allocation*—concentrating evaluations on high-potential paths. Future work could incorporate **step-level reward models** (Xia et al., 2025; Lightman et al., 2023; Uesato et al., 2022; Wang et al., 2024), **outcome verification models** (Yu et al., 2024), or learned value functions to enable denser feedback and earlier credit assignment, reducing reliance on expensive full rollouts.

**Workflow Representation.** We adopt a linear (chain-structured) workflow space. For a broad class of agentic systems, the execution on a given input can often be reasonably abstracted as an (approximately) linearized sequence of decisions and tool calls (e.g., as recorded by step-by-step traces), making the chain abstraction a pragmatic first-order model. However, this abstraction does not cover scenarios involving true concurrency or asynchronous coordination (e.g., parallel tool calls, event-driven updates, or branching that induces multiple simultaneously active threads), where execution cannot be faithfully linearized without losing key structure. Extending AWR to such settings is an important direction for future work, requiring richer representations (e.g., typed DAGs) and search procedures that balance structural expressivity with interpretability.

**Evaluation and Scope.** Our metric measures output-level structural/semantic similarity between reconstructed and

target artifacts, and is intentionally agnostic to unobservable internal traces in strict black-box settings. We therefore use SFE as a scalable *proxy* for behavioral similarity on the observed I/O distribution, rather than a guarantee of execution-level correctness.

Our evaluation focuses on code-centric domains, enabling rigorous SFE analysis. Many agentic tasks naturally produce code (data pipelines, automation, API orchestration), making this scope relevant. Extending AWR to purely natural language domains (e.g., creative writing, open-ended dialogue) would require new evaluation approaches—combining embedding-based semantic similarity with human judgment or LLM-as-judge protocols (Zheng et al., 2023). More broadly, AWR faces intrinsic limits when targets rely on proprietary tools or hidden capabilities unavailable to the reconstruction space; establishing theoretical bounds on reconstructibility remains an open problem.

## 5. Related Work

**LLM-based Agents and Search.** Recent work has increasingly focused on LLM-based agents that perform reasoning, planning, and tool use (Wei et al., 2022; Yao et al., 2023a; Hao et al., 2023; Wang et al., 2023a; Schick et al., 2023; Qin et al., 2023). A common theme is to treat agent decision-making as search: Tree of Thoughts (ToT) (Yao et al., 2023a) generalizes Chain-of-Thought (CoT) into a tree, while RAP (Hao et al., 2023) and LATS (Zhou et al., 2024) integrate MCTS with LLMs to guide multi-step reasoning. These methods primarily focus on *instance-level* search for solving a single problem instance, whereas our work targets *workflow-level reconstruction*: identifying a generic, interpretable workflow structure that matches a black-box agent system across a dataset. Concurrent search-based agents primarily optimize search dynamics for a fixed problem instance; our setting instead recovers reusable workflow structure from observed outputs.

**LLM-based Multi-Agent Systems and Automated Design.** Multi-agent frameworks such as CAMEL (Li et al., 2023a), ChatDev (Qian et al., 2024), and MetaGPT (Hong et al., 2023) coordinate multiple agents through natural language to divide roles and collaboratively solve complex tasks. Other systems extend this paradigm with code-first orchestration (Qiao et al., 2023), autonomous task agents (XAgent Team, 2023), and graph-based agent pipelines built on LangGraph (Wang & Duan, 2024). Beyond manually designed systems, a growing body of work explores automated construction and optimization of multi-agent systems, including search-based methods like AFlow (Zhang et al., 2025a) and ADAS (Hu et al., 2025), as well as optimization-based methods like DyLAN (Liu et al., 2024b) and Agent-Prune (Zhang et al., 2024). A complementary direction studies dynamic orchestration, where the coordination policy itself is learned and adapted online (Dang et al., 2025). In contrast, these approaches typically assume access to a configurable (white-box) agent architecture to optimize performance, whereas we study the complementary problem of reconstructing an explicit workflow from black-box input-output observations, applicable to both multi-agent and complex single-agent settings.

**Interpreting Agentic Behaviors.** Another related line aims to explain or interpret agent behaviors and decisions (Domenech i Vila et al., 2024; Gimenez-Abalos et al., 2024). These methods provide valuable insights into agent actions, but they do not explicitly reconstruct an executable, interpretable workflow structure of the underlying system, which is the goal of AWR. By producing an explicit surrogate, AWR complements this line by enabling not only post-hoc inspection but also direct edit, replay, and debugging of the recovered workflow.

## 6. Conclusion

We introduced AWR, a task that recovers an editable, interpretable white-box *surrogate* workflow for a black-box *agentic system* using only input–output observations. We proposed AgentXRay, which formulates reconstruction as a combinatorial search over a *Unified Primitive Space* and solves it with MCTS enhanced by *Red-Black Pruning*. Across five diverse domains spanning both multi-agent frameworks and proprietary assistants, AgentXRay achieves higher *proxy fidelity* (Static Functional Equivalence) than behavior cloning (SFT) and an unpruned MCTS baseline (AFlow), while reducing practical search cost under a fixed budget. We discuss limitations and future directions (e.g., graph-structured workflows and stronger evaluators); our results suggest that meaningful, editable workflow structure can be recovered from black-box behavior, providing a practical step toward more transparent, controllable, and debuggable agentic systems.

## Acknowledgements

This work was supported by the Shanghai Municipal Special Program for Basic Research on General AI Foundation Models (Grant No. 2025SHZDZX026D04), in collaboration with Shanghai Artificial Intelligence Laboratory.

## Impact Statement

This paper presents work whose goal is to advance the field of machine learning by improving the interpretability of multi-agent systems. While such techniques may indirectly influence downstream applications of agent-based systems, we do not foresee significant negative societal or ethical risks arising directly from this work.

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
