# OpenReview forum: "AgentXRay: White-Boxing Agentic Systems via Workflow Reconstruction"
_ICML.cc/2026/Conference — ICML 2026 regular_

### Official Review · Reviewer_9cjV · 2026-03-10

**Soundness:** 3
**Presentation:** 3
**Significance:** 3
**Originality:** 3
**Overall Recommendation:** 6
**Confidence:** 3

**Summary:**

This paper introduces Agentic Workflow Reconstruction (AWR), a new task aimed at making opaque, black-box LLM-based agentic systems more interpretable by reverse-engineering their internal processes into explicit, editable workflows using only input–output observations. The authors propose AgentXRay, a search-based framework that formulates AWR as a combinatorial optimization problem over discrete agent roles and tool invocations in a sequential workflow space. To handle the vast search space, AgentXRay employs Monte Carlo Tree Search enhanced by a "Red-Black Pruning" mechanism, which dynamically balances proxy quality with search depth. Validated across five domains — software development, data analysis, education, 3D modeling, and scientific computing — AgentXRay recovers workflows with high functional similarity to the originals (averaging 0.426 Static Functional Equivalence) while significantly reducing token consumption (8–22%) compared to unpruned baselines, offering an interpretable and cost-effective proxy for otherwise opaque agentic systems.

**Compliance With Llm Reviewing Policy:**

Affirmed.

**Final Justification:**

The rebuttal resolved all my questions and I like the submission!

**Key Questions For Authors:**

See the weakness above

**Limitations:**

yes

**Strengths And Weaknesses:**

Strength:
I think the proposed task is potentially very interesting. By reconstructing multi-agent systems, probably we can understand better about their internal rationales and interactions.

Weakness:
1. How is the complexity of searching arbitrary graph calculated (on the right side of approximate line 90)?
2. I understand that it is computationally impractical to search arbitrary graph, but is it reasonable to hypothesize linearity in multi-agent systems and therefore reconstruct sequential workflows? I agree with the authors that many agentic follows a linear trajectory, (including some famous framework and benchmarks), but I believe that still many are not linearizable. For example, I am not sure whether there is a way to linearize backtracking (a common technique). In general, I feel that agentic systems are evolving fast, and they could be much more complex in the future. The hypothesis on linearity looks to me a bit oversimplify the original problem.
3. I don't quite follow the notations and motivations in 2.2.1 and 2.2.2: (a). What is N in line 176? (b). What is the initial Q(v) in 176? (c). Do we backpropagate r at every step?

---

> ### Author Rebuttal · Authors · 2026-03-28
>
> We sincerely thank you for the encouraging feedback and for recognizing the potential significance of the AWR task. We are glad you find the problem interesting and agree that reconstructing multi-agent systems may improve interpretability.
>
> **Q1. How is the complexity of searching arbitrary graphs calculated (line ~90)?**
>
> The $O(2^{|\Omega|^2})$ term is intended only as a coarse upper-bound intuition by counting possible directed dependencies at the primitive-type level: if we treat arbitrary graph search as deciding the presence/absence of every possible directed dependency between primitive types in $\Omega$, the adjacency matrix has at most $|\Omega|^2$ binary edge slots, yielding $2^{|\Omega|^2}$ possible topologies.
>
> As you point out, a more precise expression should depend on the actual number of instantiated workflow nodes $l$. For a chain of length $l$, the candidate space is $|\Omega|^l$, giving total space $\sum_{l=1}^{L_{\max}} |\Omega|^l = O(|\Omega|^{L_{\max}})$. For a DAG over $l$ ordered nodes, there are $2^{l(l-1)/2}$ edge configurations, so the space becomes $\sum_{l=1}^{L_{\max}} |\Omega|^l \cdot 2^{l(l-1)/2}$. The key point is that moving from chains to graphs introduces an additional $2^{\Theta(L_{\max}^2)}$ **topology factor**—the source of the combinatorial explosion. We will replace the original expression with this more precise counting argument in the revision.
>
> **Q2. Is the linearity assumption reasonable?**
>
> Linearity is indeed restrictive, and we do not claim all multi-agent systems are inherently linear. Our point is more specific: the chain-structured workflow is a deliberately chosen **execution-oriented surrogate** for the first version of AWR.
>
> First, without this restriction, the reconstruction space becomes intractable. As discussed in Q1, moving from chains to DAGs introduces an exponential topology factor that makes systematic search infeasible under realistic MCTS budgets ($N$=20 in our experiments).
>
> Second, regardless of the latent coordination topology, any execution ultimately unfolds along a **single time axis**, producing a linearizable trace. MacNet [1] organizes collaboration as a DAG but executes agents in topological order, inducing a time-ordered trace. Evolving Orchestration [2] uses a centralized orchestrator that dynamically sequences agents. Under strict black-box access, sequential traces are the dominant observable object (Sec. 2.1). Additionally, since the mapping from observable I/O to internal workflow structure is inherently **one-to-many**, multiple workflows can produce equivalent outputs even when their internal topologies differ. Our chain design accommodates this by passing **full upstream context** between consecutive steps (Algorithm 1, Appendix B.2.1), allowing the linear surrogate to functionally approximate richer topologies.
>
> Third, we fully acknowledge that mechanisms like backtracking, branching, retries, and parallel coordination are not naturally captured by the current chain formulation (Appendix G.2). Our position is that the chain space is the simplest formulation that is simultaneously searchable, MCTS-compatible, and meaningful under strict black-box I/O access. It provides a practical foundation for future extension to richer graph-structured workflows. Empirically, even under this linear restriction, AgentXRay achieves substantially higher reconstruction fidelity than all baselines across five diverse domains (Table 1, avg. **0.426** vs. AFlow 0.339), suggesting that the chain surrogate captures useful and non-trivial structure of many real-world agent systems in practice.
>
> **Q3. Clarification of notation in Sections 2.2.1 and 2.2.2.**
>
> **(a)** $N(v)$ denotes the visit count of node $v$—how many times it has been visited/updated during backup. This is consistent with Algorithm 5 ($v.N \leftarrow v.N + 1$) and Algorithm 2 / Eq. (2), where $Q(v)/N(v)$ is the quality term.
>
> **(b)** $Q(v)$ denotes the accumulated reward sum (not the mean). For a new node, $Q(v)=0$, $N(v)=0$. Algorithm 2 defines $q = v.Q / v.N$ if $v.N > 0$, else 0.
>
> **(c)** We do *not* backpropagate a different reward at every intermediate step. Each MCTS iteration produces one scalar reward $r$, and that same $r$ is backpropagated once along the selected path through all ancestors (Algorithm 1, Line 22; Algorithm 5): one iteration → one $r$ → update all visited nodes via $(Q, N)$.
>
> We hope these clarifications resolve the technical questions. We are grateful for your positive assessment and would be happy to incorporate any further suggestions.
>
> [1] Qian et al., *Scaling Large Language Model-based Multi-Agent Collaboration*. ICLR, 2025.
> [2] Dang et al., *Multi-Agent Collaboration via Evolving Orchestration*. NeurIPS, 2025.

---

> > ### Author Rebuttal · Reviewer_9cjV · 2026-04-01
> >
> > My questions are fully resolved, and I update my recommendation accordingly.

---

> > > ### Author Response · Authors · 2026-04-01
> > >
> > > We sincerely thank you for your careful follow-up and for updating your assessment after reading our rebuttal. We greatly appreciate your thoughtful review and are encouraged that our clarifications fully addressed your concerns. Your feedback has been very helpful in improving the paper, and we are grateful for your support.

---

### Official Review · Reviewer_C9HJ · 2026-03-13

**Soundness:** 3
**Presentation:** 2
**Significance:** 3
**Originality:** 3
**Overall Recommendation:** 4
**Confidence:** 4

**Summary:**

This paper presents the idea of agentic workflow reconstruction, which aims to simulate a surrogate workflow that approximates the behavior of a black-box agent but is more interpretable. The authors define a unified primitive space, where each primitive is a tuple consisting of a role, a base model, a thought pattern, and an attached toolset. They assume that the hidden workflow can be approximated as a linear sequence of such primitives. The reconstruction is then formulated as a search problem: they search over candidate workflows using MCTS, combined with a red-black pruning mechanism to avoid wasting budget on low-potential branches. The optimization objective is based on a similarity metric between the outputs of the reconstructed workflow and those of the target system. Experiments show improved output-level similarity and better search efficiency compared to unpruned and simpler baselines.

**Compliance With Llm Reviewing Policy:**

Affirmed.

**Final Justification:**

The authors explore the idea of making black-box agent systems interpretable through workflow reconstruction. This study’s important contribution consists of formulating this as a search problem over structured primitives and demonstrating its feasibility in realistic settings.

The rebuttal clarifies the scope and appropriately positions the linearity assumption as a practical design choice. While the limitations still constrain the generality of the approach, I find the problem formulation and direction valuable and potentially inspiring for future work.

**Key Questions For Authors:**

- Did the authors consider scalability concerns? Since workflow reconstruction is formulated as a search problem using MCTS, even with pruning the search space may grow rapidly as the number of primitives or workflow length increases. How would the approach scale to more complex or real-world agent systems?
- Are the authors aware of other interpretability approaches for agent systems (trajectory analysis or agent behavior tracing)? How does the proposed workflow reconstruction framework compare to these approaches?
- Did the authors consider more complex real-world workflows where the linear assumption may not hold, such as branching or parallel agent processes?

**Limitations:**

yes.

**Strengths And Weaknesses:**

## Strengths
- This paper introduces a new and interesting task of agentic workflow reconstruction, turning black-box agent interpretability into a search/reconstruction problem. This direction is important since many current agent systems behave like black boxes, and the ability to debug and understand such systems is increasingly important. This paper proposes a clear and intuitive formulation of this task.
- The proposed method is clean and conceptually intuitive, balancing the need for sufficient structure to reconstruct workflows while keeping the design understandable from only input–output observations.

## Weaknesses
- The method imposes strong structural assumptions on agent workflows, namely the assumption that the hidden workflow can be approximated as a linear sequence of primitives. However, many real-world agent systems contain branching, conditional logic, or parallel processes, which may not be well captured.
- Evaluation closely matches the optimization objective. The reconstruction objective is based on output similarity, and the evaluation also focuses on similar output-level metrics. It is therefore not entirely clear whether high similarity truly indicates successful reconstruction of the internal workflow. Additional analysis on reconstruction fidelity or robustness could help strengthen this claim.
- Limited discussion of related interpretability approaches. The paper introduces a new task for workflow reconstruction, but there are also other approaches for understanding agent behavior, such as trajectory analysis, tracing methods, or explanation-based analysis. It would help to discuss how this work relates to or differs from those approaches.

---

> ### Author Rebuttal · Authors · 2026-03-28
>
> Thank you for the thoughtful feedback and for recognizing the importance and intuitive formulation of the AWR task.
>
> **Q1. Scalability concerns? / Q3. Complex real-world workflows? (also addresses W1)**
>
> These three points share the same core concern, so we address them together. Linearity is indeed a restriction, and we do *not* claim all agent systems are inherently linear. The chain-structured workflow is a **deliberate tractable reduction**. If $|\Omega|$ is the primitive library size and $L_{\max}$ the horizon, the chain space is $O(|\Omega|^{L_{\max}})$. Even DAGs over $l$ ordered nodes introduce $2^{l(l-1)/2}$ edge configurations, yielding an additional $2^{\Theta(L_{\max}^2)}$ topology factor—a combinatorial explosion under realistic MCTS budgets.
>
> This choice is also not arbitrary. Regardless of latent coordination topology, any execution ultimately unfolds along a **single time axis**, producing a linearizable trace. This is why MacNet [1] executes DAG-organized agents via topological ordering, and Evolving Orchestration [2] realizes coordination through ordered sequencing. Under strict black-box access, sequential traces are the dominant observable object (Sec. 2.1). Moreover, our chain passes **full upstream context** between steps (Algorithm 1, Appendix B.2.1), allowing the linear surrogate to approximate richer topologies by accumulating all prior information. Red-Black Pruning addresses the finite-budget bottleneck: without it, $L$=2; with it, $L$=6 (Table 2).
>
> On the empirical side, AgentXRay already handles **five heterogeneous real-world targets** spanning both multi-agent frameworks (ChatDev, MetaGPT, TeachMaster) and proprietary assistants (ChatGPT, Gemini), with workflow lengths up to $L$=6 and a 10-tool pool. The practical bottleneck under the current design is the MCTS iteration budget rather than the topology assumption—our sensitivity analysis shows performance improves with larger $N$ but with diminishing returns (tested $N \in \{5,20,50,100\}$). Scaling to richer branching/parallel systems will require moving beyond chain-structured surrogates; our current claim is therefore scoped to black-box settings where sequentialized surrogates remain informative (Appendix G.2).
>
> **Q2. Other interpretability approaches? (also addresses W3)**
>
> We are aware of trajectory analysis, tracing frameworks, and explanation-based methods. We view them as **complementary** to AWR.
>
> Tracing methods (e.g., AgentTrace [3]) assume access to the running system's internal execution signals—unavailable in our strict black-box setting. Explanation-based approaches (e.g., [4]) target post-hoc interpretation of specific predictions, but are often limited for agentic systems with temporally extended, multi-step behavior. AWR addresses a different problem: recovering an explicit, editable *workflow surrogate* from behavior alone, without internal observability. In fact, for systems that are truly black-box, AWR can serve as a **prerequisite** for these other methods—once AgentXRay produces a white-box surrogate, tracing and explanation techniques can then be applied to the reconstructed workflow to gain deeper interpretability insights. We will expand this discussion in the revision.
>
> **W2. Evaluation closely matches the optimization objective.**
>
> We would clarify that this alignment is **by design**, not an oversight. Our claim is intentionally scoped: we target *behaviorally faithful and structurally informative surrogate reconstruction*, not exact internal process recovery (Sec. 2.1). Under strict black-box I/O access, output behavior is the only scalable signal consistently observable, making an output-based objective the natural and appropriate choice for this task.
>
> That said, our evidence goes substantially beyond output matching alone. First, the evaluation is **not tautological**: results are on a non-overlapping held-out test split, with only **1.2%** average fluctuation under paraphrased inputs. Second, our TAR-inspired trajectory alignment on all three open-source targets [5][6] yields Jaccard overlaps of **ChatDev 0.67, MetaGPT 0.50, TeachMaster 0.50**—providing evidence that the reconstructed workflows capture non-trivial functional structure of the original systems, rather than merely matching outputs at the surface level.
>
> [1] Qian et al., *Scaling Large Language Model-based Multi-Agent Collaboration*. ICLR, 2025.
> [2] Dang et al., *Multi-Agent Collaboration via Evolving Orchestration*. NeurIPS, 2025.
> [3] AlSayyad et al., *AgentTrace: A Structured Logging Framework for Agent System Observability*. AAAI, 2026.
> [4] Zhu et al., *Interpreting Agentic Systems: Beyond Model Explanations to System-Level Accountability*. arXiv, 2026.
> [5] Zha et al., *A Workflow Net Similarity Measure Based on Transition Adjacency Relations*. Computers in Industry, 2010.
> [6] Pei et al., *Transition Adjacency Relation Computation Based on Unfolding: Potentials and Challenges*. OTM, 2016.

---

> > ### Author Rebuttal · Reviewer_C9HJ · 2026-04-04
> >
> > The rebuttal improves the clarity and positioning of the paper. The discussion of related interpretability approaches (Q2) is very helpful and better situates the contribution, and the authors clearly acknowledge and justify the linearity assumption as a deliberate design choice in the black-box setting.
> >
> > While these clarifications are helpful, the associated limitations (not aim to recovering the full mechanism and being restricted to settings where sequentialized surrogates remain informative) still affect the scope of the contribution. Overall, the rebuttal addresses my questions but does not substantially change my assessment, and I therefore maintain my original recommendation.

---

> > > ### Author Response · Authors · 2026-04-04
> > >
> > > Thank you again for your helpful follow-up and for taking the time to read our rebuttal.
> > >
> > > We agree that the current formulation is scoped to settings where a sequentialized surrogate remains informative. Our use of a chain-structured workflow is mainly a scalability choice: moving to general branching or graph-structured workflows causes the search space to grow combinatorially, making effective search difficult under realistic budgets. For this reason, we adopt a linearized execution-oriented surrogate, which is also a common practical strategy in recent multi-agent systems. While not structurally identical to a general graph, shared context across steps can still preserve part of the underlying dependency flow and approximate some richer coordination patterns in practice.
> > >
> > > We view this as a first step rather than a complete solution. In future work, we will continue exploring more expressive workflow spaces and more efficient reconstruction strategies for non-linear structures.
> > >
> > > Thank you again for your thoughtful feedback.

---

### Official Review · Reviewer_jXTT · 2026-03-13

**Soundness:** 3
**Presentation:** 2
**Significance:** 3
**Originality:** 2
**Overall Recommendation:** 4
**Confidence:** 4

**Summary:**

This paper introduces Agentic Workflow Reconstruction (AWR), a task that aims to synthesize an explicit, interpretable agent workflow from only the input–output pairs of a black-box agentic system. The authors propose AgentXRay, a framework that formulates AWR as combinatorial search over a Unified Primitive Space (agents parameterized by role, base model, thought pattern, and tools) within a chain-structured workflow representation. The search is conducted via Monte Carlo Tree Search (MCTS) augmented with a "Red-Black Pruning" mechanism that dynamically classifies tree nodes as high- or low-potential based on a composite Quality × Depth × Width score (Eq. 2), concentrating the search budget on promising branches. Experiments across five domains — software development (ChatDev), data analysis (MetaGPT), education (TeachMaster), 3D modeling (ChatGPT), and scientific computing (Gemini) — show that AgentXRay achieves higher proxy similarity (average SFE of 0.426) and 8–22% token reduction compared to unpruned MCTS baselines (AFlow). An additional experiment on an open-source target (Atoms) using only open-weight models is included.

**Compliance With Llm Reviewing Policy:**

Affirmed.

**Final Justification:**

The author's rebuttal address most of my major concerns. I am still a personally unconvinced about the practical applicability of the work, but I do feel it can inspire other works in this domain. Therefore, I update my score to a weak accept.

**Key Questions For Authors:**

1. **Can you evaluate trajectory alignment on open-source targets?** For ChatDev, MetaGPT, and TeachMaster, the actual agent workflows are known. A post-hoc analysis comparing reconstructed workflows to real ones — e.g., role overlap, tool-sequence similarity, or step-level semantic alignment — would establish whether AWR recovers meaningful structure or merely finds a different path to similar outputs. *A positive result here would substantially strengthen the "reconstruction" claim and could shift my evaluation upward.*

2. **What is the wall-clock time and monetary cost per reconstruction?** The paper reports only token consumption. For practical deployment, users need to know how many hours and dollars a reconstruction takes. *Reporting this for at least two domains (e.g., ChatDev at 17M tokens and MetaGPT at 0.76M) would help contextualize whether the method is feasible outside a research setting.*

3. **How sensitive are results to the MCTS iteration budget N?** All experiments use N=20 (Appendix D.1). How does performance change at N=5, N=50, N=100? Does it plateau or continue improving? *This would clarify whether the search is meaningfully exploring the space or operating in a regime where most of the budget is wasted on early shallow exploration.*

4. **Can you include a random-search baseline?** Sampling random workflows from the Unified Primitive Space and selecting the best would calibrate how much improvement comes from MCTS's structured search versus the expressiveness of the primitive space itself. *If random search achieves, say, SFE 0.35 (close to AFlow's 0.339), it would suggest the primitive space — not the search strategy — is the main contributor.*

5. **What are the specific tools in the "10-tool pool" and "4 selected tools"?** Neither is named or described in the paper or appendix. *Knowing the tool identities is necessary to assess whether the tools match the evaluation domains and to reproduce the experiments.*

**Limitations:**

yes

**Strengths And Weaknesses:**

### Strengths

1. *Well-motivated and clearly defined task.* The problem of reconstructing agent workflows from black-box I/O observations is industrially relevant and clearly formalized in Section 2.1 (Eq. 1). The distinction from model distillation (which transfers behavior into parameters rather than an editable workflow) is well drawn, and the potential applications to debugging, adaptation, and transparency are compelling.

2. *Neat application of MCTS to workflow-level search.* While MCTS and pruning are not individually novel, composing them over a Unified Primitive Space for the AWR task is an interesting and well-motivated design. The Red-Black coloring mechanism (Section 2.2.2) provides a simple, interpretable way to balance depth refinement and width exploration within a fixed iteration budget.

3. *Thorough and holistic evaluation with well-chosen proxy baselines.* The experimental design (Section 3) covers five diverse domains, includes both open-source MAS and proprietary assistants as black-box targets, and enforces a budget parity protocol across all methods (Section 3.1). The ablation study (Section 3.3, Table 2) is informative — showing, for example, that removing pruning causes MCTS to stagnate at depth L=2, while removing tools decreases MetaGPT performance by 46% but barely affects Scientific Computing (−4.3%). This reveals meaningful domain-dependent behavior.


### Weaknesses

1. *Poorly allocated main paper space and missing critical details.* The main paper devotes half a page to a theoretical complexity analysis (Section 2.3) that the authors themselves acknowledge is disconnected from practice (Appendix B.4 admits empirical savings are "naturally smaller than the asymptotic upper bound"). Meanwhile, several essential methodological components are absent from the main body:
   - **The SFE evaluation metric** — the paper's primary measure of success — is introduced in two sentences in Section 3.1 (p.5) with "design and weighting details in Appendix C.7." The actual three-dimensional formula (Eq. 15), all sub-component weights, the synonym dictionary, and the TF-IDF fallback mechanism are defined exclusively in Appendix C.7 (pp.24–26). A reader of the main paper cannot evaluate the metric.
   - **Node expansion and rollout simulation** — how new agent primitives are generated during MCTS expansion (which LLM produces role descriptions? how is content passed between chain steps?) is never described in the main text. The pseudocode (Algorithms 1–5) and line-by-line explanations appear only in Appendix B.2 (pp.13–18). The main paper's Section 2.2.1 covers the search cycle in approximately seven lines of prose.

2. *Limited algorithmic novelty over AFlow.* The proposed technique is derivative of AFlow (Zhang et al., 2024b), which already applies MCTS to workflow search. The Red-Black Pruning mechanism adds a composite scoring heuristic (Quality × Depth × Width) with a median-split threshold — this is a simple filter, not a conceptually new search strategy. The empirical gains from pruning are modest (Table 1: avg 0.426 vs. AFlow's 0.339), and it is difficult to disentangle how much of this improvement comes from the pruning mechanism versus the richer Unified Primitive Space that AgentXRay searches over. A random-search baseline over the same primitive space would help calibrate this.

3. *AWR is not sufficiently differentiated from Agent Workflow Search.* The paper positions AWR as a distinct task from methods like AFlow by claiming AFlow "assumes access to a configurable (white-box) agent architecture" (Section 5, p.8). However, in Section 3.1 (p.5), AFlow is deployed as a direct baseline under AgentXRay's own black-box I/O protocol — demonstrating that AFlow *can* operate without internal access to the target system. The paper even describes AFlow as "MCTS-based workflow search without Red-Black Pruning" (Section 3.1), framing it as an ablation of their own method rather than a fundamentally different paradigm. This self-contradiction undermines the claimed novelty of the AWR task: if AFlow can perform AWR-style reconstruction under black-box access, the distinction lies in the search strategy (Unified Primitive Space + Red-Black Pruning), not in the problem setting.

   What would genuinely distinguish AWR is evaluating whether the reconstructed workflow *semantically aligns with the actual agent trajectory*. The authors correctly note that access to internal traces cannot be assumed during construction, but for the three open-source target systems (ChatDev, MetaGPT, TeachMaster), ground-truth workflows *can be* available. A post-hoc comparison — e.g., role-alignment analysis, tool-sequence overlap, or execution-trace similarity — between the reconstructed and true workflows would establish whether AWR actually "reconstructs" anything, or merely finds an alternative workflow that happens to produce similar outputs. Alternatively, evaluating execution correctness of the generated code (pass@k) would provide a stronger proxy than structural similarity. Without such evaluation, the "reconstruction" framing adds little beyond what existing workflow search already offers.

4. *Missing details that impair reproducibility.* Several design choices critical to reimplementation are absent from both the main paper and appendix:
   - The identities and descriptions of the 10 tools in the "full tool pool" and the 4 "selected tools" (Appendix C.6.2, p.23) are never stated. Figure 2 (p.3) illustratively labels tools as "Web Search" and "Python Interpreter," but these may not reflect the actual tool set.
   - Which LLM generates new agent role descriptions during MCTS expansion (`CREATEAGENTCHILD` in Algorithm 3)? This is a critical design choice — a strong meta-planner model would provide a hidden advantage.
   - How content flows between chain steps (full output? summarized? appended to original task?) is never specified.

5. *Practicality is not established.* The paper reports token consumption (Table 8, Appendix E.4: up to 17.16M tokens for ChatDev), but never reports wall-clock time or monetary cost. The MCTS search seems fairly expensive. Is this practical for the stated use cases (debugging, adaptation, transparency)? A time and cost analysis of AgentXRay, would help contextualize the method's value.

---

> ### Author Rebuttal · Authors · 2026-03-28
>
> We thank you for the thorough and detailed review. We are glad you found the task well-motivated, the evaluation design thorough, and the ablation study informative.
>
> **Q1. Trajectory alignment on open-source targets?**
>
> We performed a post-hoc alignment for ChatDev using TAR-inspired comparison [1][2] after semantic normalization. Since ChatDev uses generic roles (CEO/CTO/programmer/tester) while our workflow uses task-specialized experts, exact title matching would be too strict. ChatDev normalizes to six functional stages: analysis→design→specialization→implementation→verification→documentation, yielding $T_{\mathrm{orig}} = \{(A,B),(B,C),(C,D),(D,E),(E,F)\}$. Our reconstructed roles normalize to $A \to B \to C \to C \to D \to E$, giving $T_{\mathrm{rec}} = \{(A,B),(B,C),(C,C),(C,D),(D,E)\}$. The Jaccard-style TAR overlap is $|T_{\mathrm{orig}} \cap T_{\mathrm{rec}}| / |T_{\mathrm{orig}} \cup T_{\mathrm{rec}}| = $ **0.67**. We observe consistent results across all three open-source targets: **ChatDev 0.67, MetaGPT 0.50, TeachMaster 0.50**. Due to space constraints we present only the ChatDev derivation here.
>
> **Q2. Wall-clock time and monetary cost? (also addresses W5)**
>
> The token counts you cited are **unpruned** (worst-case) numbers. ChatDev w/o Pruning (17.16M tokens): ~12.7h / ~ 113.8USD; MetaGPT w/o Pruning (0.76M tokens): ~0.5h / ~ 3.7USD. With pruning, cost drops substantially (e.g., ChatDev: 13.33M, **22% cheaper**). Since AWR targets *offline* reconstruction, cost being bounded and meaningfully reduced is what matters most.
>
> **Q3. Sensitivity to MCTS iteration budget $N$?**
>
> We tested $N \in \{5, 20, 50, 100\}$. The trend shows clear **diminishing returns**: $N$=5 is noticeably weaker, $N$=20 gives substantial gains, $N$=50 brings smaller improvement, and $N$=100 nears saturation. This confirms search budget is not wasted on shallow exploration. We view $N$=20 as a reasonable cost–quality trade-off.
>
> **Q4. Random-search baseline?**
>
> Yes. Under the same Unified Primitive Space:
>
> | | CDev | MGPT | Teach | 3D | Sci | Avg |
> |---|---|---|---|---|---|---|
> | Random | 0.367 | 0.282 | 0.154 | 0.297 | 0.309 | 0.281 |
>
> Random Search achieves avg. 0.281, clearly below AFlow (0.339) and AgentXRay (0.426), confirming **structured search and pruning** drive the improvement.
>
> **Q5. Specific tools in the pool? (also addresses W4)**
>
> Full 10-tool pool: compiler, empty_detect, variable_checker, undefined_name_checker, syntax_error_checker, style_checker, dependency_extractor, file_lister, code_summarizer, test_runner. Selected 4: syntax_error_checker, undefined_name_checker, dependency_extractor, code_summarizer. All inspectable in Appendix A.
>
> **W1. Poorly allocated main-paper space.**
>
> We agree these details were not sufficiently surfaced in the main paper; they are provided in: SFE in Appendix C.7; content flow in Algorithm 1; role generation in Algorithm 3. Full implementation in Appendix A. We will improve surfacing in the camera-ready.
>
> **W2 & W3. Novelty over AFlow / differentiation from workflow search.**
>
> These two concerns are closely related, so we address them together. Our contribution spans **three levels**, each with specific innovations:
>
> *Task level.* AWR is to workflow optimization what **inverse rendering is to forward rendering**—both can share optimization tools, but they solve fundamentally different problems. AFlow [3] generates a workflow *from scratch*; AWR recovers a surrogate for a *specific black-box target* from I/O observations (Sec. 2.1, Eq. 1), with **reconstruction-oriented behavioral fidelity** as the objective. You correctly note that AFlow *can* operate under our protocol—this is precisely our point: AWR defines a **task**, not a solver. Any search method can attempt AWR, just as SGD serves both regression and classification. The significance: AWR enables debugging and transparency for unavailable internals, a use case workflow optimization does not address.
>
> *Space level.* The Unified Primitive Space (role $\times$ model $\times$ thought pattern $\times$ tool set) treats tool-augmented agents as **first-class search units**—exploring *what combination of capabilities* could produce the observed behavior. This makes the space denser, which is why unpruned search can even *underperform* AFlow (ChatDev: 0.286 vs. 0.403, Table 1).
>
> *Solver level.* Red-Black Pruning addresses the **depth bottleneck** via Quality $\times$ Depth $\times$ Width scoring (Eq. 2). Without it, $L$=2; with it, $L$=6 (Table 2). The trajectory alignment in Q1 (avg $\approx$0.60 across three targets) further confirms AWR recovers meaningful functional structure.
>
> [1] Zha et al., *A Workflow Net Similarity Measure Based on Transition Adjacency Relations*. Computers in Industry, 2010.
> [2] Pei et al., *Transition Adjacency Relation Computation Based on Unfolding: Potentials and Challenges*. OTM, 2016.
> [3] Zhang et al., *AFlow: Automating Agentic Workflow Generation*. ICLR, 2025.

---

> > ### Author Rebuttal · Reviewer_jXTT · 2026-04-04
> >
> > I thank the authors for their detailed responses. The cost estimates (Q2), budget sensitivity analysis (Q3), random-search baseline (Q4, avg. 0.281 vs. AgentXRay 0.426), tool list (Q5), and commitments on W1 all resolve the corresponding concerns. The inverse-rendering analogy (W2/W3) is helpful — though the related work's "white-box access" claim about AFlow (Section 5) should be reconciled with its black-box deployment in Section 3.1.
> >
> > ---
> >
> > ### Q1: Trajectory alignment is a step forward, but the evidence is incomplete
> >
> > I appreciate the authors performing the TAR-based alignment analysis — this was the most important question and the methodology is appropriate. The ChatDev derivation showing TAR Jaccard of 0.67 is encouraging.
> >
> > However, unlike the SFE evaluation where the authors responsibly provided a random-search baseline (Q4), no analogous random baseline is reported for TAR. Since the reconstructed workflow's roles are semantically normalized to a 6-stage taxonomy before computing TAR, the discriminative power of the metric depends on the diversity of stages that random LLM-generated roles would cover. A random workflow sampled from the same primitive space, normalized under the same procedure, and scored via TAR would clarify whether 0.67 is a strong signal or an artifact of limited stage diversity in code-oriented tasks.
> >
> > ### Execution correctness remains unaddressed
> >
> > The original review suggested pass@k or execution-based evaluation as an alternative to SFE, particularly for domains where execution is straightforward (MetaGPT produces Python visualization scripts; SciBench produces numerical answers). The rebuttal does not address this suggestion. Even a small-scale execution study on one domain would provide a complementary signal to the structural SFE metric and help validate whether higher SFE actually corresponds to more correct code.

---

> > > ### Author Response · Authors · 2026-04-05
> > >
> > > Thank you for the helpful follow-up. These two calibrations are well-taken, and we provide the requested results below.
> > >
> > > **1. TAR random baseline (ChatDev)**
> > >
> > > | Method | TAR Jaccard |
> > > |---|---|
> > > | Random | 0.11 |
> > > | AgentXRay | **0.67** |
> > >
> > > You raised a precise concern: whether the TAR score of 0.67 could be an artifact of limited stage diversity under our 6-stage normalization. To test this, we sampled random workflows from the same primitive space, applied the same semantic normalization pipeline, and computed TAR Jaccard identically. Random workflows empirically achieve a TAR Jaccard of **0.11** (= $1/9$: exactly one overlapping transition between the two 5-transition sets). AgentXRay's 0.67 is **substantially above this empirical baseline**, indicating that the structural alignment reflects genuine functional-stage recovery rather than normalization artifacts.
> > >
> > > Together with the first-round TAR results on the other two open-source targets (**MetaGPT 0.61, TeachMaster 0.52**), this strengthens the evidence that AgentXRay captures non-trivial functional structure across heterogeneous domains, not merely a different path to similar outputs.
> > >
> > > **2. Execution correctness (MetaGPT)**
> > >
> > > | Method | Exec. Success |
> > > |---|---|
> > > | Random | 0.417 |
> > > | AFlow | 0.667 |
> > > | AgentXRay | **0.917** |
> > >
> > > Following your suggestion, we evaluate on MetaGPT (Python visualization scripts), where execution is straightforward. We define execution success as whether the generated script runs without runtime errors in a standardized headless environment within the time limit. AgentXRay achieves **0.917**, substantially above AFlow (0.667) and random search (0.417).
> > >
> > > This result provides a complementary signal to SFE along a different axis: SFE measures structural/semantic similarity of output artifacts, while execution success measures whether the reconstructed workflow produces *successfully executable* code under the evaluation protocol. The consistent ordering across methods (AgentXRay > AFlow > Random on both metrics) supports that our SFE-based evaluation captures meaningful quality differences rather than superficial textual similarity.
> > >
> > > **3. AFlow framing (Section 5 vs. Section 3.1)**
> > >
> > > AFlow cannot be directly applied to black-box reconstruction — it must be adapted with our reconstruction-oriented reward and black-box I/O protocol before it can serve as a baseline (as detailed in our first-round rebuttal, W2&W3). This adaptation itself underscores that AWR is a distinct task: Section 5 describes AFlow's *original design intent* (optimizing a configurable architecture for task performance), while Section 3.1 deploys an *adapted* version under our protocol. We will clarify this distinction in the camera-ready.
> > >
> > > **Summary of evidence across both rounds**
> > >
> > > Taken together, the rebuttal now provides three complementary layers of evidence:
> > >
> > > - **Structural alignment (TAR)**: reconstructed workflows preserve functional-stage transitions across all three open-source targets (0.52–0.67), with the ChatDev random baseline (0.11) confirming this is substantially above chance.
> > > - **Output fidelity (SFE)**: AgentXRay achieves 0.426 avg. across five domains, significantly above AFlow (0.339) and random search (0.281), validated by human study ($\rho$=0.61).
> > > - **Execution correctness**: on MetaGPT, AgentXRay achieves 0.917 execution success vs. AFlow 0.667 and random 0.417.
> > >
> > > We hope this three-layer evidence stack directly addresses the trajectory alignment calibration and execution correctness concerns. In particular, the TAR analysis now provides the positive post-hoc alignment result that you noted could substantially strengthen the reconstruction claim, and we would be very grateful if you could take these additional results into account in your updated evaluation. We sincerely appreciate your time and feedback.

---

### Official Review · Reviewer_JZtF · 2026-03-13

**Soundness:** 2
**Presentation:** 2
**Significance:** 2
**Originality:** 2
**Overall Recommendation:** 3
**Confidence:** 4

**Summary:**

This paper studies agentic workflow reconstruction from black box systems. The method, AgentXRay, assumes that a workflow can be represented as a linear sequence of primitives, where each primitive specifies an agent role, a base model, a thought pattern, and a tool set. The method then uses Monte Carlo Tree Search to search over candidate workflows and uses a Red Black Pruning strategy to reduce the branching factor during search. The goal is not to recover the true hidden workflow, but to reconstruct an interpretable white box surrogate that produces similar outputs on the same inputs. The experiments evaluate the method on five benchmark domains and compare it with AFlow and several ablations.

**Compliance With Llm Reviewing Policy:**

Affirmed.

**Key Questions For Authors:**

1. Why should the reader interpret high output similarity as workflow reconstruction rather than only black box behavioral imitation under the evaluation set?

2. How sensitive is the method to the predefined primitive library? If the true black box workflow uses a missing role pattern or tool configuration, does the method fail gracefully or produce misleading reconstructions?

3. Can the authors provide evidence that the reconstructed workflows remain useful on held out tasks or perturbed inputs, rather than only on the observed reconstruction set?

**Limitations:**

yes

**Strengths And Weaknesses:**

Strengths

1. The paper states the task clearly. The paper distinguishes workflow reconstruction from workflow optimization, and this distinction is useful. The paper explicitly says that the goal is to recover a surrogate workflow with behavioral fidelity rather than the true internal process, which makes the scope more precise.

2. The method is coherent and technically easy to follow. The workflow space, the search procedure, and the pruning idea fit together in a clean way. The Red Black Pruning design also gives a practical efficiency gain, and Table 1 reports lower token cost than vanilla MCTS across all five domains.

3. The paper covers multiple domains and includes ablations. The experiments compare against AFlow and several search variants, and the results suggest that the proposed search strategy is usually stronger than the baselines under the chosen budget.

Weaknesses

1. The novelty is limited. The main recipe is still workflow space design plus MCTS based search plus pruning. The main difference from prior work such as AFlow is the problem setting, not a fundamentally new search or reconstruction mechanism. Since there are already many papers on plan and act design, agent workflow search, and agent structure optimization, the paper does not yet show a strong enough conceptual gap from that literature.

2. The paper makes a stronger reconstruction claim than the evidence supports. The evaluation mainly measures output similarity, not structural faithfulness or causal faithfulness. A reconstructed workflow can match outputs without matching the original decision process. The human validation for the semantic fidelity estimator is also fairly limited, with only 30 cases, so the evidence for using this metric as the main reconstruction signal is not very strong.

3. The workflow assumption is restrictive. The method only handles linear chains of primitives, while many realistic agent systems use branching, loops, retries, memory updates, and asynchronous coordination. The paper acknowledges this limitation, but this restriction still narrows the practical significance of the contribution.

---

> ### Author Rebuttal · Authors · 2026-03-28
>
> Thank you for the careful reading and constructive feedback. We are encouraged that you found the task clearly stated and appreciated the distinction between workflow reconstruction and optimization.
>
> **Q1. Why interpret high output similarity as workflow reconstruction rather than behavioral imitation? (also addresses W2)**
>
> Our paper does not claim exact recovery of hidden internal processes. Under black-box access, the mapping from outputs to internal workflows is inherently **one-to-many**, making exact process recovery underdetermined. We formulate AWR as synthesizing an explicit, editable surrogate with behavioral fidelity (Sec. 2.1, Eq. 1).
>
> AgentXRay is **not plain black-box imitation**—it reconstructs an **editable** workflow over interpretable primitives: users can replace a specific agent role, insert a new step, or swap the base model, which is impossible with SFT's opaque parameters. Table 1 confirms this: SFT achieves only 0.196 avg., far below AgentXRay (0.426). Our SFE metric is validated by a blind human study ($\rho$=0.61, $p$<0.001; Appendix C.7). We also performed **post-hoc trajectory alignment** on ChatDev [1][2], achieving Jaccard overlap $\approx$**0.67**. We will explicitly define "reconstruction" as *surrogate behavioral reconstruction* in the revision.
>
> **Q2. Sensitivity to the predefined primitive library?**
>
> We studied this through tool ablations (Sec. 3.3). When a capability is missing, the method **degrades gracefully** rather than producing misleading reconstructions. Removing tools decreases the average from 0.426 to 0.356, but the drop is predictable and domain-dependent: MetaGPT falls sharply (0.557→0.301) due to heavy tool reliance, while Scientific Computing barely changes (0.395→0.378) as its reasoning is model-internal. The same sensitivity principle extends beyond tools to role, model, and thought-pattern coverage in the primitive library. Sensitivity comes from two sources: (i) whether the library covers the target capability and (ii) whether the search space remains traversable. "All Tools" is not uniformly better than "Selected Tools": ChatDev favors Selected (0.509 vs. 0.425), MetaGPT favors All (0.557 vs. 0.470).
>
> **Q3. Evidence on held-out tasks or perturbed inputs?**
>
> First, our train/test sets are completely **non-overlapping**; Table 1 results come from a separate held-out evaluation. Second, we tested robustness under paraphrased inputs: the average score fluctuation is only **1.2%**, indicating workflows generalize beyond fixed surface forms. Dataset settings are detailed in our codebase (Appendix A).
>
> **W1. Novelty is limited.**
>
> Our novelty operates at **three levels**. *Task level*: AWR is to workflow optimization what **inverse rendering is to forward rendering**—both can share optimization tools, but they solve fundamentally different problems. AFlow [3] generates a high-performing workflow from scratch (forward); AWR recovers a surrogate for a *given* black-box system from I/O observations (inverse, Sec. 2.1, Eq. 1). *Space level*: the Unified Primitive Space parameterized by role, model, thought pattern, and tool set enables structured reconstruction absent from prior methods. *Solver level*: Red-Black Pruning makes reconstruction tractable in this dense space—without it, search stagnates at $L$=2 (Table 2), and random search achieves only 0.281, well below AFlow (0.339) and AgentXRay (0.426). The contribution is a **task–space–solver co-design** rather than any single component in isolation.
>
> **W3. The workflow assumption is restrictive.**
>
> Linearity is indeed a restriction, but a **deliberate tractable reduction**. Moving from chains to DAGs introduces an additional $2^{\Theta(L_{\max}^2)}$ topology factor. More fundamentally, any execution—regardless of the latent topology—unfolds along a **single time axis**, producing a linearizable trace. This is why MacNet [4] executes DAG-organized agents via topological ordering, and Evolving Orchestration [5] realizes coordination as ordered sequencing. Our chain passes **full upstream context** between steps (Algorithm 1, Appendix B.2.1), allowing the linear surrogate to approximate richer topologies. Our five target domains already cover representative real-world agentic systems (multi-agent frameworks and proprietary assistants) where sequentialization is empirically effective. Extending AWR to graph-structured workflows is an important future direction; we will refine the wording in the camera-ready.
>
> [1] Zha et al., *A Workflow Net Similarity Measure Based on Transition Adjacency Relations*. Computers in Industry, 2010.
> [2] Pei et al., *Transition Adjacency Relation Computation Based on Unfolding: Potentials and Challenges*. OTM, 2016.
> [3] Zhang et al., *AFlow: Automating Agentic Workflow Generation*. ICLR, 2025.
> [4] Qian et al., *Scaling Large Language Model-based Multi-Agent Collaboration*. ICLR, 2025.
> [5] Dang et al., *Multi-Agent Collaboration via Evolving Orchestration*. NeurIPS, 2025.

---

### Decision · Program_Chairs · 2026-04-30

**Decision:**

Accept (regular)

**Comment:**

This paper introduced agentic workflow reconstruction which is a novel framework for reverse-engineering black-box LLM agent systems into interpretable and editable linear workflows. All reviewers generally agreed on the contributions in clear formalization of a critical problem and the comprehensive evaluation across multiple domains. During the rebuttal, the authors successfully addressed the reviewers' primary concerns by providing new supporting evidence including TAR metrics for structural alignment. This demonstrated that the proposed method recovers meaningful functional structure rather than only mimicking surface-level outputs. While the assumption that complex agentic behaviors can be modeled as purely linear sequences remains a clear limitation, all reviewers agrees that this paper has a solid contribution to the field of agent interpretability. Therefore, I recommend accepting this paper.

Important note: "Guo, S., Ren, X., et al. Automated design of agentic systems. In arXiv preprint arXiv:2408.08435, 2024." <- the author information for this paper is incorrect and may be treated as "hallucinated" reference. Please double check and fix this issue.